# Narrowing Information Bottleneck Theory for Multimodal Image-Text Representations Interpretability

**Zhiyu Zhu**[1]**, Zhibo Jin**[1]**, Jiayu Zhang**[2]**, Nan Yang**[3]**, Jiahao Huang**[3] **,
Jianlong Zhou**[1] **& Fang Chen**[1] [*]

University of Technology Sydney[1], SuZhouYierqi[2], University of Sydney[3]

## Abstract

The task of identifying multimodal image-text representations has garnered increasing attention, particularly with models such as CLIP (Contrastive Language-Image Pretraining), which demonstrate exceptional performance in learning complex associations between images and text. Despite these advancements, ensuring the interpretability of such models is paramount for their safe deployment in real-world applications, such as healthcare. While numerous interpretability methods have been developed for unimodal tasks, these approaches often fail to transfer effectively to multimodal contexts due to inherent differences in the representation structures. Bottleneck methods, well-established in information theory, have been applied to enhance CLIP's interpretability. However, they are often hindered by strong assumptions or intrinsic randomness. To overcome these challenges, we propose the Narrowing Information Bottleneck Theory, a novel framework that fundamentally redefines the traditional bottleneck approach. This theory is specifically designed to satisfy contemporary attribution axioms, providing a more robust and reliable solution for improving the interpretability of multimodal models. In our experiments, compared to state-of-the-art methods, our approach enhances image interpretability by an average of 9%, text interpretability by an average of 58.83%, and accelerates processing speed by 63.95%. Our code is publicly accessible at `https://github.com/LMBTough/NIB`.

## 1 Introduction

CLIP (Contrastive Language-Image Pretraining) has rapidly become a pivotal model in the field of multimodal learning, especially excelling in its ability to connect the visual and textual modalities (Lin et al., 2023). By training on large-scale image-text pairs collected from the internet, CLIP is capable of performing zero-shot classification and image-text retrieval tasks, making it an indispensable component of modern generative artificial intelligence (Novack et al., 2023; Jiang & Ye, 2023). With its strong visual-textual understanding capabilities, CLIP can generate, classify, and explain content without the need for fine-tuning, providing robust support for various generative AI applications. As one of the most representative Multimodal Image-Text Representation (MITR) methods, CLIP's core strength lies in mapping images and texts into a shared embedding space, significantly enhancing the performance of multimodal tasks.

Despite its outstanding performance in MITR tasks, developing effective interpretability methods to reveal CLIP's decision-making mechanisms has become increasingly important. The black-box nature of CLIP's multimodal embeddings presents significant challenges in high-risk applications such as medical diagnosis and content moderation, where transparency and reliability are crucial (Eslami et al., 2021; Tong et al., 2024; Yuan et al., 2024; Zhu et al., 2024b). A deeper understanding of how CLIP establishes associations between visual and textual representations is essential to ensure the transparency and trustworthiness of its outputs.

There have been numerous interpretability methods focused on unimodal tasks Ribeiro et al. (2016); Sundararajan et al. (2017); Zhu et al. (2024a), but these methods are not designed for the unique

---

[*]Corresponding author: fang.chen@uts.edu.au

characteristics of MITR tasks, resulting in suboptimal performance when directly applied to such tasks. However, there are existing interpretability methods specifically developed for MITR tasks. Despite their development, these methods often suffer from randomness issues, requiring additional sampling or loss information (Wang et al., 2023), which leads to a crisis of trust in the interpretability method itself. These issues will be further analyzed in the related work section.

Given that CLIP can generate unique image and text representations without the need for additional samples and can directly establish their correlations, it is possible to design an interpretability algorithm that unveils the mechanisms behind these correlations without requiring extra sampling. M2IB (Wang et al., 2023), based on the Information Bottleneck Principle (IBP), proposes an interpretability method that does not require additional samples. This method controls the amount of feature information through a Bottleneck layer and optimizes the parameters of this layer to maximize the mutual information between the representations and the task target while minimizing the correlation between the representations and the original sample. Although IBP has a solid theoretical foundation in information theory, in practice, its reliance on hyperparameters and random sampling often introduces bias into the interpretation results. We will discuss this issue in detail in Section 3.2.

To address the aforementioned challenges, we propose a novel Narrowing Information Bottleneck Theory (NIBT). Through rigorous theoretical derivation, NIBT effectively eliminates the randomness and hyperparameter dependency in IBP, resulting in more deterministic interpretability outcomes. Additionally, we introduce a new concept of negative property, which identifies feature dimensions that negatively impact the model's predictions, further enhancing the model's interpretability. Our Contributions as follows:

- We systematically summarize existing MITR interpretability methods and highlight the limitations.
- We propose and derive the novel Narrowing Information Bottleneck Theory, which enables interpretation of MITR tasks without randomness, while preserving the advantages of the IBP.
- Our research significantly improves the interpretability of the CLIP model, and we release our method as open-source for further research and application.

## 2 RELATED WORK

### 2.1 CONTRASTIVE LANGUAGE-IMAGE PRETRAINING (CLIP)

Radford et al. (2021) introduced CLIP, which learns multimodal embeddings of images and text by training image and text encoders on large-scale image-text paired data. This enables CLIP to establish connections between the two modalities within a unified embedding space, facilitating zero-shot transfer, where the model can make predictions based on natural language descriptions without relying on task-specific labeled data. However, the complexity of these multimodal tasks necessitates a focus on interpretability to ensure that the model's decisions are grounded in meaningful features. Studying the interpretability of CLIP helps verify whether the model genuinely understands the relationship between vision and language, as opposed to relying on spurious correlations in the data.

### 2.2 TRADITIONAL INTERPRETABILITY METHODS

Traditional interpretability methods for deep learning models were initially designed for unimodal tasks. Early methods, such as Saliency Maps, generate fine-grained heatmaps by computing the gradient of the model's output with respect to input pixels. However, these methods are sensitive to noise and often yield coarse explanations. Grad-CAM (Selvaraju et al., 2017), by computing the gradient of activation maps in convolutional layers, produces class-specific heatmaps, making the explanations more intuitive, particularly for convolutional neural networks (CNNs). RISE (Petsiuk et al., 2018) further advances the field by introducing a black-box method that applies random masks to different regions of the input image, observes changes in the model's output, and generates heatmaps that account for both global and local interpretability. RISE's advantage lies in its model-agnostic nature, making it applicable to any architecture. However, due to its reliance on

random sampling, it is computationally expensive and may introduce some noise. LIME Ribeiro et al. (2016), another black-box method, perturbs the input locally and trains a surrogate model to provide locally linear explanations, making it applicable to any model, though it can sometimes produce inaccurate explanations in complex tasks. Overall, these interpretability methods designed for unimodal tasks do not perform well in multimodal tasks, as we demonstrate in our experiments.

With the introduction of the Sensitivity Axiom and Implementation Invariance Axiom by Sundarara-jan et al. (2017), point-wise interpretable methods have rapidly evolved. The Sensitivity Axiom requires the sensitivity of a model's output to align with its attribution values, while the Implementation Invariance Axiom demands that functionally equivalent models yield the same attribution results, regardless of implementation. Currently, the most advanced attribution methods based on adversarial attacks Jin et al. (2024), such as AGI (Pan et al., 2021) and MFABA (Zhu et al., 2024c), satisfy both axioms and have shown strong interpretability for traditional CNN models. However, these methods have not been optimized for multimodal tasks, and directly modifying their loss functions for multimodal tasks is infeasible. Moreover, they are primarily designed for unimodal tasks, rely on downstream tasks for explanations, and lack adaptation to multimodal contexts. As a result, while traditional interpretability methods have made progress in unimodal tasks, there remains a significant gap in addressing multimodal tasks and novel models like CLIP, requiring further optimization and extension.

## 2.3 INTERPRETABILITY METHODS FOR MULTIMODAL TASKS

Currently, existing interpretability methods for multimodal tasks still exhibit several limitations that require improvement, as shown in Table 1. In the following sections, we will provide a detailed explanation of the causes and effects of these limitations.

**No Extra Example** indicates that no additional samples are required during the interpretation process, which is crucial because in real-world scenarios, we do not know what samples to select, nor can we explain why a particular pair of samples are correlated. For instance, if we aim to explain which parts of an image depict a cat, the image in the CLIP model already exhibits high activation with respect to the text *cat*. Therefore, we should not need to reference 100 additional images of cats and 100 images without cats. A well-trained model that already understands the semantics should not require such sampling. **No Randomness** means that the calculation process involves no randomness, as randomness reduces trust in the interpretability method. **No Specific Structure** means that the method does not depend on a particular model structure. **No Info Loss** ensures that no information is lost during the interpretation process, such as interpreting only a subset of the model's output. **Current Model** indicates that the method explains the model as it currently exists, without constructing a new model for interpretation. **No Downstream Task** means that no downstream tasks are required for the explanation process.

Table 1: Comparison of interpretability methods based on several criteria: whether they require no extra examples, no randomness, need no specific structure, avoid information loss, explain the current model, and don't rely on downstream tasks.

| Method | No Extra Example | No Randomness | No Specific Structure | No Info Loss | Current Model | No Downstream Task |
|---|---|---|---|---|---|---|
| COCOA | ○ | ○ | ● | ○ | ● | ● |
| TEXTSPAN | ○ | ○ | ○ | ○ | ● | ● |
| Hossain et al. | ○ | ○ | ● | ○ | ● | ● |
| LICO | ○ | ○ | ● | ○ | ○ | ○ |
| FALCON | ○ | ○ | ● | ● | ● | ● |
| M2IB | ● | ○ | ● | ○ | ● | ● |
| NIB (Ours) | ● | ● | ● | ● | ● | ● |

M2IB (Wang et al., 2023) introduced a multimodal information bottleneck method aimed at explaining the decision-making process of vision-language pre-trained models by compressing task-irrelevant information to highlight key predictive features. However, this approach introduces additional complexity, which will be analyzed further when discussing the IBP theory. Similarly, COCOA (Lin et al., 2022) extended Integrated Gradients (IG) to multimodal tasks by incorporating positive and negative sample pairs in its loss function, but this requires sampling additional relevant examples, introducing extraneous information that may not be directly relevant to explaining the current sample.

Other methods like TEXTSPAN (Gandelsman et al., 2023) and Hossain et al. also suffer from sample dependency. TEXTSPAN requires constructing a specific text set to calculate similarity with the image, limiting its scope to predefined sets, while Hossain et al. relies on selecting training data samples based on L2 distance in the embedding space, which is not always feasible in practical settings. LICO (Lei et al., 2024) attempts to create an interpretable model by retraining it to maintain feature relationships between text and image, but this results in explaining the newly trained model rather than the original one, and randomness is introduced through batch sampling. FALCON (Kalibhat et al., 2023) explains each dimension in the feature space by finding images that highly activate a specific feature, but this approach does not provide explanations for individual samples, limiting its applicability. Overall, many of these methods face challenges such as reliance on additional samples, randomness, or structural dependencies, making them less suitable for clear, direct explanations of pre-trained models.

## 3 PRELIMINARY

### 3.1 PROBLEM DEFINITION

Following the setup of CLIP (Radford et al., 2021), a trained MITR model can be defined as follows: let $f_I : \mathbb{R}^n \to \mathbb{R}^d$ denote the image encoder, which transforms an input image $x_I \in \mathbb{R}^n$ into a $d$-dimensional image representation; $f_T : \mathbb{R}^m \to \mathbb{R}^d$ denote the text encoder, which transforms an input text $x_T \in \mathbb{R}^m$ into a $d$-dimensional text representation. We can use $\cos \langle f_I(x_I), f_T(x_T) \rangle$ to evaluate the matching performance between the visual and textual modalities. Additionally, the representations can be directly applied to downstream tasks (Sanghi et al., 2022; Zhou et al., 2023). In the following, we use $f$ to represent either $f_I$ or $f_T$. By substituting $f$ with $f_I$, we obtain the results associated with the image modality, and similarly, we can derive the results for the text modality.

For an $L$-layer neural network, we can decompose it into the concatenation of two neural networks $f^{1-l} \circ f^{l-L}(x)$ at the $l$-th layer. For ease of expression, we use $z = f^{1-l}(x)$ to represent the latent feature of the intermediate layer.

Our goal is to construct an interpretability method $A$ that yields $A(x) \in \mathbb{R}^{|x|}$. The larger the value of $A$, the more important that dimension is for the representation.

### 3.2 THE INFORMATION BOTTLENECK PRINCIPLE

The information bottleneck principle (Tishby et al., 2000), based on information theory, introduces the bottleneck to control the amount of information passing through it, aiming to find the minimal feature encoding that retains the least amount of information from the original sample while preserving the necessary information for a given task. For ease of reading and understanding, we provide a brief explanation and simplify the notation, with more detailed analysis provided in **Appendix A**. The goal of the information bottleneck principle is to construct an optimization function and find the optimal parameter $\lambda$:

$$\lambda^* = \max_\lambda I(\tilde{z}, Y) - \beta I(\tilde{z}, x; \lambda) \tag{1}$$

where $I(x, Y) = H(x) - H(x|Y) = H(Y) - H(Y|x)$ represents the mutual information between events $x$ and $Y$, which can be interpreted as the reduction in uncertainty about event $x$ after observing event $Y$. Intuitively, the stronger the correlation between the two, the greater the reduction in uncertainty, and thus the larger the mutual information. Here, $x$ represents the input sample, $\tilde{z}$ represents the encoding of $x$, which can be understood as the extracted features, and $Y$ represents the given task. $\lambda$ controls the size of the bottleneck. We emphasize the **Key Point 1**: $\lambda^*$ represents the value of $\lambda$ when the mutual information between the encoding and the task is maximized while minimizing the correlation with the original sample (i.e., extracting as few features as possible). The optimization process follows (Schulz et al., 2020).

## 4 METHOD

In this section, we first deconstruct how the Information Bottleneck Principle extracts the importance distribution of sample features and analyze the shortcomings of applying this theory to the

interpretability of deep learning. We then introduce our Narrowing Bottleneck Theory, which is rigorously derived and applied to the interpretability of Multimodal Image-Text Representations.

## 4.1 ANALYSIS OF THE INFORMATION BOTTLENECK PRINCIPLE (IBP)

Several works (Wang et al., 2023; Schulz et al., 2020) have applied the IBP to the interpretability of neural networks. Their approach typically involves inserting a Bottleneck layer at the $l$-th layer of the neural network, with $\lambda \in \mathbb{R}^{|z|}$ controlling the amount of information in the $l$-th layer feature $z = f^{1-l}(x)$. The optimal solution for $\lambda$, based on Equation 1, is found through gradient descent iterations, and the control is achieved as follows:

$$\tilde{z}_{ic}(\lambda) = \lambda_{ic} \cdot z_{ic} + (1 - \lambda)\varepsilon, \quad \varepsilon \sim N(\mu, \sigma^2) \tag{2}$$

Here, following the Grad-CAM approach (Selvaraju et al., 2017), the dimension of $z$ is split into two parts for discussion: $i$ represents the spatial encoding, and $c$ represents the channel encoding (for instance, $z \in \mathbb{R}^{w \times h \times c}$, where $w$, $h$, and $c$ represent width, height, and channels, respectively. The $w \times h$ portion is simplified as $i$). The noise distribution follows $\varepsilon \sim N(\mu, \sigma^2)$, and any distribution independent of $z$ can be used, but a normal distribution is chosen for computational simplicity. The parameters $\sigma^2$ and $\mu$ can be arbitrarily specified. When $\lambda = \lambda^*$, the importance of $z_i$ is given by $\sum_c D_{KL}(P(\tilde{z}_{ic}(\lambda)|x)\|N(\mu, \sigma^2))$. Intuitively, this measures the uncertainty allowed in the $i$-th feature dimension under the condition of the **Key Point 1**. If this dimension is very close to an independent noise distribution of sample $x$, a small KL divergence implies irrelevance to $x$, i.e., unimportance, and vice versa.

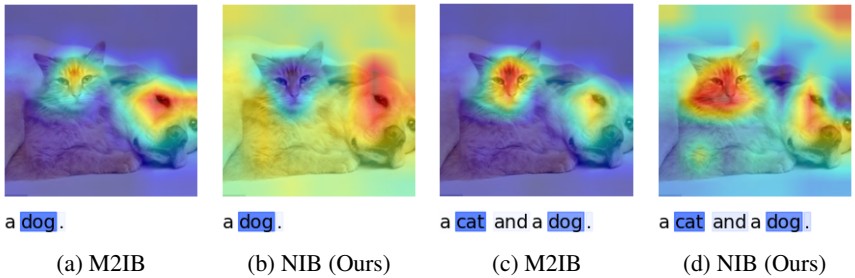

|  a dog . | a dog . | a cat and a dog . | a cat and a dog . |
| :---: | :---: | :---: | :---: |
| (a) M2IB | (b) NIB (Ours) | (c) M2IB | (d) NIB (Ours) |

Figure 1: Illustrations of feature attributions produced by different methods.

While the IBP theory itself is solid, its application faces several limitations as following:

1. the optimization process introduces randomness, as the calculation of $I(\tilde{z}, Y)$ depends on $\tilde{z}_{ic}(\lambda)$, which involves sampling noise $\lambda$. This results in numerous local optima in the optimization of $\lambda_{ic}$, leading to variations between runs.

2. the hyperparameter $\beta$ in the optimization objective significantly influences the interpretability results (Wang et al., 2023). $\beta$ controls the trade-off between the two mutual information terms, allowing different task information to result in entirely different explanations.

3. the KL divergence is always positive, preventing the explanation from reflecting negative properties (As shown in Figure 1, our proposed method successfully distinguishes and excludes negative properties from the explanation. In Figure 1d, the M2IB method continues to highlight irrelevant negative features, such as the cat's face, even when the subject is a dog. However, in Figure 1b, our method correctly ignores these negative properties, focusing on more relevant, positive features, showcasing its improved attribution performance.)

4. the explanation results do not directly reflect the association between feature dimensions and $I(\tilde{z}, Y)$ (the explanation is derived by optimizing Equation 1 to obtain $\lambda$, followed by computation).

To address the three above issues, we propose the Narrowing Bottleneck Theory.

## 4.2 THE NARROWING INFORMATION BOTTLENECK THEORY

In this section, we introduce three core theorems of the Narrowing Bottleneck Theory and propose our Narrowing Information Bottleneck Method (NIB) algorithm based on them.

We continue to introduce a Bottleneck layer at the $l$-th layer and use $\lambda$ to control the information flow, while the scalar $\lambda$ serves as a universal update parameter for each layer, the flow of information for each feature dimension is determined independently. More Details of $\lambda$ please refer to Appendix G

$$\tilde{z}_{ic}(\lambda) = \lambda \cdot z_{ic} + \varepsilon, \quad \varepsilon \sim N(0, \sigma^2) \tag{3}$$

However, unlike Equation 3, we use the same scalar $\lambda \in \mathbb{R}$ to control all dimensions of $z$. Additionally, we assume that the noise follows a distribution with zero mean and variance $\sigma^2$, and we eliminate the noise weighting factor $1 - \lambda$ (as long as the noise distribution is independent of $z$, this modification does not affect the bottleneck's properties). $z_{ic}$ is deterministically obtained by the model, ensuring there is no inherent randomness in its computation.

For simplicity, we present an equivalent form, where $\mathbb{I}$ represents the identity matrix:

$$\tilde{z}(\lambda) = \lambda \cdot z + \varepsilon, \quad \varepsilon \sim N(0, \sigma^2 \mathbb{I}) \tag{4}$$

**Theorem 1** (Narrowing Information Bottleneck). *Given* $0 \leq \lambda_1 < \lambda_2 \leq 1$*, we have* $\sup I(\tilde{z}(\lambda_1), x) < \sup I(\tilde{z}(\lambda_2), x)$*, and when* $\lambda = 0$*, we have* $I(\tilde{z}(0), x) = 0$*.*

*Proof.* We start by expressing the mutual information $I(\tilde{z}, x)$ as:

$$\begin{aligned} I(\tilde{z}, x) &= E_x \left[ D_{\text{KL}} \left[ P(\tilde{z} \mid x) \| Q(\tilde{z}) \right] - D_{\text{KL}} \left[ P(\tilde{z}) \| Q(\tilde{z}) \right] \right] \\ &\leq E_x \left[ D_{\text{KL}} \left[ P(\tilde{z} \mid x) \| \tilde{Q}(\tilde{z}) \right] \right] \end{aligned} \tag{5}$$

Given that $P(\tilde{z}(\lambda) \mid x) = N(\lambda_z, \sigma^2 \mathbb{I})$, we can compute the difference between the mutual information at two values of $\lambda$ as follows:

$$\sup I(\tilde{z}(\lambda_1), x) - \sup I(\tilde{z}(\lambda_2), x) = E_x \left[ \frac{1}{2} \cdot \frac{1}{\sigma^2} \left( \lambda_1^2 - \lambda_2^2 \right) \| \mu \|^2 \right] \tag{6}$$

Since $(\lambda_1^2 - \lambda_2^2) < 0$, it follows that:

$$\sup I(\tilde{z}(\lambda_1), x) < \sup I(\tilde{z}(\lambda_2), x) \tag{7}$$

This completes the key proof. Further details can be found in **Appendix B**. $\square$

**Theorem 1 shows that by decreasing the value of $\lambda$, we can reduce the mutual information between $x$ and its encoding, and when $\lambda = 1$, $I(\tilde{z}(\lambda), x)$ reaches its maximum value.** Since the computation involves noise sampling, we aim to remove the randomness caused by Theorem 1, which leads us to Theorem 2.

**Theorem 2.** *When* $\sigma^2 \to 0$*, given* $0 \leq \lambda_1 < \lambda_2 \leq 1$*, we have:*

$$\sup_{\sigma^2 \to 0} I(\tilde{z}(\lambda_1), x) < \sup_{\sigma^2 \to 0} I(\tilde{z}(\lambda_2), x) \tag{8}$$

In Theorem 2, we demonstrate that the conclusion of Theorem 1 holds as $\sigma^2$ tends to zero. Specifically, as $\sigma^2 \to 0$, $z(\lambda)$ converges to $\lambda z$. In practical scenarios, due to precision limitations, the two expressions will become indistinguishable, effectively eliminating any inherent randomness.

**Theorem 3.** *Given the function* $I(\tilde{z}, Y)$*, the following holds:*

$$\sum_i \sum_c \int_0^1 \frac{\partial I(\tilde{z}(\lambda), Y)}{\partial \tilde{z}_{ic}(\lambda)} \frac{\partial \tilde{z}_{ic}(\lambda)}{\partial \lambda} d\lambda = I(\tilde{z}(1), Y) - I(\tilde{z}(0), Y) \tag{9}$$

Building on Theorem 2, we can adjust the value of $\lambda$ to control the size of $I(\tilde{z}(\lambda), x)$. When $\lambda = 0$, $I(\tilde{z}(\lambda), x)$ is minimized, and when $\lambda = 1$, $I(\tilde{z}(\lambda), x)$ is maximized. Therefore, $I(\tilde{z}(\lambda), x)$ can be

viewed as a function of $\lambda$, where the process from 1 to 0 corresponds to the bottleneck transitioning from fully open to completely closed. The importance $A(z_i)$ of $z_i$ can be expressed as:

$$A(z_i) = \sum_c \int_0^1 \frac{\partial I(\tilde{z}(\lambda), Y)}{\partial \tilde{z}_{ic}(\lambda)} \frac{\partial \tilde{z}_{ic}(\lambda)}{\partial \lambda} d\lambda \tag{10}$$

**The total importance across all dimensions of $z$ equals the loss in $I(\tilde{z}(\lambda), x)$ caused by narrowing the bottleneck from fully open to closed.** Negative values are also allowed, as some features may reduce $I(\tilde{z}(\lambda), x)$. This process eliminates the need for balancing two mutual information terms, thus avoiding the introduction of the $\beta$ hyperparameter and preventing instability in the interpretability results.

For the design of $I(\tilde{z}(\lambda), x)$, we follow the work of Wang et al. (2023), using $\cos \langle f_I(x_I), f_T(x_T) \rangle$ as an equivalent replacement. It is worth noting that if we aim to obtain the attribution result for the image modality, $I(\tilde{z}(\lambda), x)$ becomes $\cos \langle f^{l-L}(\lambda f^{1-l}(x_I)), f_T(x_T) \rangle$. Additionally, since $i$ in $z_i$ represents the spatial encoding corresponding to the original encoding, the importance distribution of the original sample features $A(x)$ can be obtained by performing linear interpolation on $\lambda$ from 0 to 1, as described in (Wang et al., 2023; Schulz et al., 2020). Theorem 2, Theorem 3, and the proofs of the Sensitivity and Implementation Invariance axioms are provided in the **Appendix**.

## 5 EXPERIMENTS

### 5.1 MODELS AND DATASETS

In this study, we follow the experimental setup of M2IB (Wang et al., 2023), utilizing the pre-trained CLIP model with a Vision Transformer (ViT-B/32) (Dosovitskiy, 2020) as the visual encoder. CLIP's joint optimization of image and text alignment has demonstrated outstanding performance in multimodal tasks. We conduct experiments on three different datasets: Conceptual Captions (Sharma et al., 2018), ImageNet (Deng et al., 2009), and Flickr8k (Hodosh et al., 2013). Each of these datasets has unique characteristics, providing diverse visual and textual inputs for the model. Conceptual Captions is a large-scale image-text alignment dataset containing automatically generated image-text pairs, helping the model learn a shared feature space between vision and language. ImageNet, a classic image classification dataset, contains a large number of annotated images and a wide range of class labels, making it a standard dataset for training and evaluating visual models. Flickr8k is a relatively small image-text alignment dataset consisting of 8,000 images and their corresponding natural language descriptions, commonly used to assess multimodal alignment in image captioning and text generation tasks.

### 5.2 PARAMETER SETTINGS

We reduced the number of parameters required by the IBP-based method while retaining the core hyperparameters used during the generation of saliency maps, including the number of iterations (*num_steps*) and the layer number.

**num_steps: Number of Iterations** The *num_steps* parameter refers to the number of iterations used during gradient optimization, and it primarily affects the precision. It determines how many updates are made to the feature maps in each layer during gradient backpropagation. A larger *num_steps* generally leads to higher precision, as the model is given more iterations to accumulate gradients and refine attribution results. However, as the number of iterations increases, so does the computational cost, necessitating a balance between accuracy and efficiency in practical applications. In our experiments, *num_steps* is set to 10, which has been experimentally verified to provide a higher precision result while maintaining relatively low computational overhead.

**layer number: Layer Number** The *layer number* refers to the identifier of the specific layer chosen from the neural network model as the bottleneck layer. In this study, we selected the 9th layer (*layer number* = 9), indicating that we extract the hidden states from the 9th layer for generating saliency maps. The choice of this layer is motivated by the fact that intermediate layers typically contain rich contextual information, reflecting both low-level features and some high-level abstract representations. Specifically, using the hidden states from the 9th layer allows us to capture the model's intermediate features, avoiding the low-level signals from early layers or the overly abstract

representations from deeper layers. The feature maps from this layer have been shown in practice to effectively support the generation of saliency maps, striking a balance between feature detail and semantic representation.

## 5.3 EVALUATION METRICS

In the evaluation of attribution algorithms, traditional *insertion score* and *deletion score* metrics rely on task-specific confidence outputs. However, as our experiments do not include task labels or confidence information, incorporating downstream task outputs would weaken the generality of the interpretability methods. Additionally, the metric ROAR+ (an Extension of ROAR (Hooker et al., 2019)), which requires retraining the model after removing key features, incurs a high computational cost, particularly when dealing with complex models and large-scale datasets, significantly increasing time and resource consumption. For these reasons, we reference two model-output-based evaluation methods proposed by Wang et al. (2023): **Confidence Drop** and **Confidence Increase** (Chattopadhay et al., 2018), to evaluate the performance of attribution algorithms.

The **Confidence Drop** and **Confidence Increase** are evaluation metrics used to assess the effectiveness of attribution methods. The former measures whether model performance decreases when less important features are removed, with the ideal scenario being that only high attribution scores are retained and the removal of other features does not significantly impact performance. A lower value of **Confidence Drop** indicates better performance of the attribution method. The latter evaluates whether removing noisy information from the input enhances the model's confidence, with the expectation that the removal of irrelevant features should increase the model's confidence. A higher value of **Confidence Increase** indicates better performance of the attribution method. Both metrics serve to gauge whether the attribution method effectively identifies and preserves important features while mitigating the impact of noise.

## 5.4 BASELINE

We compare our proposed Narrowing Information Bottleneck (NIB) method against several well-established attribution techniques to evaluate its effectiveness. The baseline methods include M2IB (Wang et al., 2023), RISE (Petsiuk et al., 2018), Grad-CAM (Selvaraju et al., 2017), the method by Chefer et al. (2021), Saliency Maps (Simonyan, 2013), MFABA (Zhu et al., 2024c), and FastIG (Hesse et al., 2021).

## 5.5 RESULT

Table 2: Performance comparison of the proposed NIB method with existing attribution methods across three datasets: Conceptual Captions, ImageNet, and Flickr8k. The evaluation metrics include Image Confidence Drop, Image Confidence Increase, Text Confidence Drop, Text Confidence Increase, and Frames Per Second (FPS). Lower confidence drop and higher confidence increase indicate better performance, while higher FPS reflects better computational efficiency. NIB consistently achieves superior performance in both accuracy and efficiency across all datasets.

| Dataset | Method | M2IB | RISE | Grad-CAM | Chefer et al. (2021) | SM | MFABA | FastIG | NIB (Ours) |
|---------|--------|------|------|----------|----------------------|-----|-------|--------|------------|
| Conceptual Captions | Img Conf Drop ↓ | 1.1171 | 1.4197 | 4.1064 | 2.0138 | 10.4351 | 10.1878 | 10.5117 | **0.9439** |
| | Img Conf Incr ↑ | 39.3 | 28.8 | 20.2 | 33.65 | 2.95 | 2.6 | 2.9 | **42.5** |
| | Text Conf Drop ↓ | 1.706 | 0.8002 | 1.7994 | 0.9333 | 1.0723 | 1.0503 | 0.9718 | **0.2705** |
| | Text Conf Incr ↑ | 37.4 | **43.95** | 34.4 | 45.3 | 40.05 | 36.25 | 41.25 | **43.95** |
| | FPS ↑ | 0.6621 | 0.1 | 1.1686 | 1.272 | 0.928 | 0.2494 | 0.9384 | **1.5817** |
| ImageNet | Img Conf Drop ↓ | 1.1615 | 1.001 | 2.5483 | 1.6636 | 4.7331 | 5.0242 | 4.7905 | **0.9012** |
| | Img Conf Incr ↑ | 49.4 | **54** | 33.9 | 44 | 16.4 | 12.7 | 16.9 | 53.1 |
| | Text Conf Drop ↓ | 2.6018 | 0.9928 | 2.6424 | 1.6732 | 1.7631 | 1.7437 | 1.6486 | **0.4193** |
| | Text Conf Incr ↑ | 25.4 | 46.8 | 25.7 | 29.9 | 33.1 | 28.5 | 34.8 | **56.1** |
| | FPS ↑ | 0.7995 | 0.1084 | 2.3115 | 2.7867 | 1.5711 | 0.2758 | 1.5384 | **2.4481** |
| Flickr8k | Img Conf Drop ↓ | 1.4731 | 3.01 | 5.1869 | 2.6214 | 12.154 | 12.07 | 12.2244 | **1.4495** |
| | Img Conf Incr ↑ | **28.1** | 5.7 | 13.6 | 26.8 | 0.1 | 0.1 | 0.1 | **28.1** |
| | Text Conf Drop ↓ | 2.0783 | 0.8914 | 2.1823 | 1.362 | 1.0797 | 1.1551 | 1.3098 | **0.4562** |
| | Text Conf Incr ↑ | 34.7 | 46.4 | 34.2 | 42.6 | 45.9 | 42.6 | 43.9 | **55.3** |
| | FPS ↑ | 0.7397 | 0.1076 | 1.958 | 2.4601 | 1.3973 | 0.2748 | 1.3944 | **2.1995** |

The performance of our proposed NIB (Narrowing Information Bottleneck) method is compared with several existing attribution methods, including M2IB, RISE, Grad-CAM, and others, across three different datasets: Conceptual Captions, ImageNet, and Flickr8k. The evaluation is based on four key metrics: Image Confidence Drop, Image Confidence Increase, Text Confidence Drop, and Text Confidence Increase. Additionally, the computational efficiency is assessed through Frames Per Second (FPS).

On the Conceptual Captions dataset, NIB demonstrates superior performance with an Image Confidence Drop of 0.9439, outperforming M2IB by 0.1732 units and Grad-CAM by 3.1625 units. Similarly, NIB achieves an Image Confidence Increase of 42.5, surpassing Grad-CAM by 22.3 units and RISE by 13.7 units, reflecting its strong ability to improve model focus by removing irrelevant features. For text-based metrics, NIB shows a notable improvement in Text Confidence Drop, with a 1.4355 unit advantage over M2IB and a 1.5289 unit gap with Grad-CAM. In terms of computational efficiency, NIB achieves the highest FPS of 1.5817, providing a substantial performance boost over RISE and other methods.

In the ImageNet dataset, NIB maintains its leading position, with the lowest Image Confidence Drop (0.9012), outperforming M2IB by 0.2603 units and Grad-CAM by 1.6471 units. Additionally, NIB achieves the highest Image Confidence Increase of 53.1, showing a 19.2 unit improvement over Grad-CAM and a 3.7 unit improvement over M2IB. For text metrics, NIB continues to excel with the lowest Text Confidence Drop (0.4193), representing a 2.1825 unit gap over M2IB and a 2.2231 unit gap over Grad-CAM. The FPS score for NIB is also competitive at 2.4481, showing high efficiency in real-time applications.

On the Flickr8k dataset, NIB achieves the lowest Image Confidence Drop (1.4495), only slightly better than M2IB by 0.0236 units, but significantly outperforms Grad-CAM by 3.7374 units. In terms of Image Confidence Increase, NIB ties with M2IB at 28.1, while exceeding Grad-CAM by 14.5 units. NIB also leads in Text Confidence Drop, with a score of 0.4562, outperforming M2IB by 1.6221 units and Grad-CAM by 1.7261 units. The computational efficiency of NIB remains high, with an FPS of 2.1995, reflecting its ability to maintain high-speed performance compared to slower methods like RISE.

In summary, the proposed NIB method consistently outperforms existing attribution techniques across all datasets, providing better attribution accuracy and computational efficiency. The improvements in both Confidence Drop and Confidence Increase metrics demonstrate NIB's capability to identify key features and remove irrelevant ones, enhancing the interpretability and robustness of multimodal models. Please see the attribution results images in the GitHub repository.

# 6 ABLATION RESULT

## 6.1 ABLATION STUDY OF *num_steps*

Table 3: Ablation study results on the *num_steps* parameter, comparing different values (5, 10, 15, and 20) across three datasets: Conceptual Captions, ImageNet, and Flickr8k. The evaluation metrics include Image Confidence Drop, Image Confidence Increase, Text Confidence Drop, and Text Confidence Increase.

| Dataset | *num_steps* | Img Conf Drop ↓ | Img Conf Incr ↑ | Text Conf Drop ↓ | Text Conf Incr ↑ |
|---|---|---|---|---|---|
| Conceptual Captions | 5 | 0.9386 | 42.4 | 0.2056 | 43.4 |
| | 10 | 0.9439 | 42.5 | 0.2705 | 43.95 |
| | 15 | 0.9424 | 42.75 | 0.3688 | 44.9 |
| | 20 | 0.9378 | 43.05 | 0.4701 | 44.3 |
| Imagenet | 5 | 0.9554 | 51.5 | 0.3164 | 56.7 |
| | 10 | 0.9012 | 53.1 | 0.4193 | 56.1 |
| | 15 | 0.9558 | 53.4 | 0.4563 | 56.4 |
| | 20 | 0.9691 | 54.3 | 0.4957 | 56.4 |
| Flickr8k | 5 | 1.4547 | 26.7 | 0.3766 | 54.1 |
| | 10 | 1.4495 | 28.1 | 0.4562 | 55.3 |
| | 15 | 1.4443 | 26.7 | 0.6222 | 53.2 |
| | 20 | 1.4504 | 27.3 | 0.8326 | 52.5 |

In the ablation study of the *num_steps* parameter, we investigate the impact of varying the number of optimization iterations on the performance of our proposed NIB method. As shown in Table 3, we conducted experiments with *num_steps* values of 5, 10, 15, Target Layer Number fixed at 9, and 20 across the three datasets.

The results indicate that as the number of steps increases, there is a trade-off between attribution accuracy and computational cost. For Conceptual Captions, increasing *num_steps* from 5 to 20 slightly improves the Text Confidence Drop but shows diminishing returns after *num_steps* = 10, with only minor improvements in accuracy but a noticeable increase in computational overhead. Similar trends are observed in ImageNet and Flickr8k, where the best performance in terms of Image and Text Confidence Drop occurs at *num_steps* = 10. Beyond this point, the gains are marginal, and the results suggest that setting *num_steps* to 10 provides an optimal balance between accuracy and efficiency.

## 6.2 ABLATION STUDY OF *target_layer*

Table 4: Ablation study results on the *target_layer* parameter, comparing layers 3, 6, and 9 across the Conceptual Captions, ImageNet, and Flickr8k datasets. The evaluation metrics include Image Confidence Drop, Image Confidence Increase, Text Confidence Drop, and Text Confidence Increase.

| Dataset | *target_layer* | Img Conf Drop $\downarrow$ | Img Conf Incr $\uparrow$ | Text Conf Drop $\downarrow$ | Text Conf Incr $\uparrow$ |
|---|---|---|---|---|---|
| Conceptual Captions | 3 | 0.8616 | 42.2 | 1.2758 | 38.7 |
| | 6 | 0.8514 | 43.55 | 0.9867 | 40.1 |
| | 9 | 0.9439 | 42.5 | 0.2705 | 43.95 |
| ImageNet | 3 | 0.6889 | 57 | 2.3727 | 31.9 |
| | 6 | 0.7793 | 56.1 | 2.4207 | 32.5 |
| | 9 | 0.9012 | 53.1 | 0.4193 | 56.1 |
| Flickr8k | 3 | 1.3022 | 26.5 | 1.5068 | 43.1 |
| | 6 | 1.2875 | 28.3 | 1.1981 | 46.9 |
| | 9 | 1.4495 | 28.1 | 0.4562 | 55.3 |

In the ablation study of the *target_layer* parameter, we explore the impact of selecting different layers for generating saliency maps. Specifically, we evaluate the performance of layers 3, 6, and 9 across the Conceptual Captions, ImageNet, and Flickr8k datasets, with *num_steps* fixed at 10.

The results in Table 4 reveal that layer 9 generally yields the best performance across all datasets. For Conceptual Captions, layer 9 achieves the lowest Text Confidence Drop (0.2705) and the highest Text Confidence Increase (43.95), indicating that the saliency maps generated from this layer provide the most accurate and interpretable attributions. Similarly, in the ImageNet dataset, layer 9 performs well, with a moderate Image Confidence Drop (0.9012) and the highest Text Confidence Increase (56.1), demonstrating that it effectively captures important features for both image and text alignment.

In contrast, selecting earlier layers (3 and 6) results in higher Confidence Drop scores, particularly in the Text Confidence Drop metric, suggesting that these layers lack the necessary high-level semantic information. Therefore, the results indicate that layer 9 strikes an optimal balance between capturing rich feature representations and providing interpretable attributions, making it the most effective choice for generating saliency maps in the NIB method.

## 7 CONCLUSION

This paper introduces the Narrowing Information Bottleneck Theory (NIBT) to address the challenges of randomness and hyperparameter sensitivity in explaining multimodal models like CLIP. By re-engineering the traditional Bottleneck method, NIBT improves interpretability for both image and text representations. The proposed method demonstrates superior performance in terms of both attribution accuracy and computational efficiency across multiple datasets. These advancements contribute to a more transparent and reliable interpretation of complex multimodal tasks, paving the way for broader applications of explainable AI in high-stakes environments.

## CODE OF ETHICS AND ETHICS STATEMENT

All authors of this paper have read and adhered to the ICLR Code of Ethics[1]during the research, development, and writing of this work. We affirm that this paper complies with all ethical guidelines outlined in the code. No part of our research involved human subjects, and there are no significant concerns regarding privacy, fairness, or potential conflicts of interest. All datasets and methodologies used are publicly available, and no sponsorship influenced the content or findings of this work.

Additionally, the interpretability methods proposed in this paper aim to improve model transparency and are intended for enhancing the trustworthiness of AI systems, especially in sensitive domains such as healthcare. We believe that our contributions will support ethical AI deployment by making models more interpretable and accountable.

## REPRODUCIBILITY STATEMENT

We have taken multiple steps to ensure the reproducibility of our results. The detailed descriptions of the models, datasets, and training protocols used in our experiments are provided in the main text. Specific hyperparameters, including the number of iterations and selected layers for saliency map generation, are also reported in Section 5.2. Furthermore, the code for implementing our proposed Narrowing Information Bottleneck Theory (NIBT) and the associated datasets are available in the Anonymous Repository[2]. These resources should enable the community to reproduce our findings and apply the methods to their own work.

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

## A    PRINCIPLES OF INFORMATION THEORY

### A.1    PROPERTIES OF MUTUAL INFORMATION

1. **Non-negativity**

$$I(X;Y) \geq 0$$

2. **Symmetry**

$$I(X;Y) = I(Y;X)$$

3. **Relationship with Conditional Entropy and Joint Entropy**

$$\begin{aligned}
I(X;Y) &= H(X) - H(X|Y) \\
&= H(Y) - H(Y|X) \\
&= H(X) + H(Y) - H(X,Y) \\
&= H(X,Y) - H(X|Y) - H(Y|X)
\end{aligned}$$

4. **Relationship with Kullback-Leibler (KL) Divergence**

$$\begin{aligned}
I(X;Y) &= \sum_y p(y) \sum_x p(x|y) \log_2 \frac{p(x|y)}{p(x)} \\
&= \sum_y p(y) D_{\mathrm{KL}}(p(x|y)\|p(x)) \\
&= \mathbb{E}_Y \left[ D_{\mathrm{KL}}(p(x|y)\|p(x)) \right]
\end{aligned}$$

## B    PROOF OF THEOREM 1

$$\begin{aligned}
\mathrm{I}[x,\tilde{z}] &= \mathbb{E}_x \left[ D_{\mathrm{KL}}[P(\tilde{z} \mid x)\|P(\tilde{z})] \right] \\
&= \int_x p(x) \left( \int_{\tilde{z}} p(\tilde{z} \mid x) \log \frac{p(\tilde{z} \mid x)}{p(\tilde{z})} d\tilde{z} \right) dx \\
&= \int_x \int_{\tilde{z}} p(x,\tilde{z}) \log \frac{p(\tilde{z} \mid x)}{p(\tilde{z})} \frac{q(\tilde{z})}{q(\tilde{z})} d\tilde{z} dx \\
&= \int_x \int_{\tilde{z}} p(x,\tilde{z}) \log \frac{p(\tilde{z} \mid x)}{q(\tilde{z})} d\tilde{z} dx + \int_x \int_{\tilde{z}} p(x,\tilde{z}) \log \frac{q(\tilde{z})}{p(\tilde{z})} d\tilde{z} dx \\
&= \int_x \int_{\tilde{z}} p(x,\tilde{z}) \log \frac{p(\tilde{z} \mid x)}{q(\tilde{z})} d\tilde{z} dx + \int_{\tilde{z}} p(\tilde{z}) \left( \int_x p(x \mid \tilde{z}) dx \right) \log \frac{q(\tilde{z})}{p(\tilde{z})} d\tilde{z} \\
&= \mathbb{E}_x \left[ D_{\mathrm{KL}}[P(\tilde{z} \mid x)\|Q(\tilde{z})] \right] - D_{\mathrm{KL}}[P(\tilde{z})\|Q(\tilde{z})] \\
&\leq \mathbb{E}_x \left[ D_{\mathrm{KL}}[P(\tilde{z} \mid x)\|Q(\tilde{z})] \right]
\end{aligned}$$

Given this, we can simplify the final result as:

$$\begin{aligned}
I(\tilde{z},x) &= E_x \left[ D_{\mathrm{KL}}\left( P(\tilde{z} \mid x)\|Q(\tilde{z}) \right) - D_{\mathrm{KL}}\left( P(\tilde{z})\|Q(\tilde{z}) \right) \right] \\
&\leq E_x \left[ D_{\mathrm{KL}}\left( P(\tilde{z} \mid x)\|\tilde{Q}(\tilde{z}) \right) \right]
\end{aligned}$$

$$\tilde{z}(\lambda) = \lambda z + \varepsilon, \quad \varepsilon \sim N(0, \sigma^2 \mathbb{I})$$

is equivalent to:

$$\tilde{z}_{ic}(\lambda) = \lambda z_{ic} + \varepsilon, \quad \varepsilon \sim N(0, \sigma^2)$$

Given that:

$$P(\tilde{z} \mid x) = N(\lambda_z, \sigma^2 \mathbb{I}),$$

we let $Q(\tilde{z}) \sim N(0, \sigma^2 \mathbb{I})$. Note that $Q(\tilde{z})$'s covariance matrix can be $\sigma^2 \mathbb{I}$ since activations after linear or convolution layers tend to follow a Gaussian distribution (Klambauer et al., 2017).

$$D_{\mathrm{KL}}(P(\tilde{z} \mid x) \| N(0, \sigma^2 \mathbb{I})) = \frac{1}{2} \left[ \mathrm{tr}\left( \Sigma^{-1} \Sigma \right) + (\mu - 0)^T \Sigma^{-1} (\mu - 0) - i \times c \right]$$

Since the covariance matrices of $P(\tilde{z} \mid x)$ and $N(0, \sigma^2 \mathbb{I})$ are the same, $\mathrm{tr}(\Sigma^{-1} \Sigma) = i \times c$, and $\log \left( \frac{\det(\Sigma)}{\det(\Sigma)} \right) = 0$, the remaining term simplifies to:

$$D_{\mathrm{KL}}(P(\tilde{z} \mid x) \| N(0, \sigma^2 \mathbb{I})) = \frac{1}{2} \cdot \frac{1}{\sigma^2} \| \mu \|^2$$

Thus, we have:

$$\sup I(\tilde{z}, x) = E_x \left[ \frac{1}{2} \cdot \frac{1}{\sigma^2} \| \mu \|^2 \right]$$

Given that $P$ and $Q$ follow normal distributions with means $\mu_p$ and $\mu_q$, and covariance matrices $\Sigma_p$ and $\Sigma_q$, respectively, we use the following KL divergence formula:

$$D_{\mathrm{KL}}(p(x) \| q(x)) = \frac{1}{2} \left[ (\mu_p - \mu_q)^\top \Sigma_q^{-1} (\mu_p - \mu_q) + \mathrm{tr}(\Sigma_q^{-1} \Sigma_p) - n \right]$$

$$\sup I(\tilde{z}(\lambda_1), x) - \sup I(\tilde{z}(\lambda_2), x) = E_x \left[ \frac{1}{2} \cdot \frac{1}{\sigma^2} \left( \lambda_1^2 - \lambda_2^2 \right) \| \mu \|^2 \right]$$

$$= E_x \left[ \frac{1}{2} \cdot \frac{1}{\sigma^2} \left( \lambda_1^2 - \lambda_2^2 \right) \| \mu \|^2 \right]$$

Since $\lambda_1, \lambda_2 \in [0, 1]$ and $\lambda_1 < \lambda_2$, we have:

$$\sup I(\tilde{z}(\lambda_1), x) < \sup I(\tilde{z}(\lambda_2), x)$$

Thus, Theorem 1 is proven. When $\lambda = 0$, we have:

$$P(\tilde{z}(0) \mid x) = N(0, \sigma^2 \mathbb{I}),$$

which is the same as $Q(\tilde{z})$, leading to:

$$D_{\mathrm{KL}}(P(\tilde{z}(0) \mid x) \| Q(\tilde{z})) = 0$$

Hence, $I(\tilde{z}(0), x) = 0$.

## C  PROOF OF THEOREM 2

When $\sigma \to 0$, we have:

$$E_x \left[ \frac{1}{2} \cdot \frac{1}{\sigma^2} \left( \lambda_1^2 - \lambda_2^2 \right) \| \mu \|^2 \right] \to 0, \quad \text{as} \quad \lambda_1^2 - \lambda_2^2 \to 0^-$$

Thus, Theorem 2 is proven.

## D  PROOF OF THEOREM 3

The proof follows the standard method of change of variables in integral calculus. Detailed steps can be found in textbooks on calculus.

# E   PROOF OF SENSITIVITY AXIOMS

$$\sum_i \sum_c \int_0^1 \frac{\partial I(\tilde{z}(\lambda), Y)}{\partial \tilde{z}_{ic}(\lambda)} \frac{\partial \tilde{z}_{ic}(\lambda)}{\partial \lambda} d\lambda = I(\tilde{z}(1), Y) - I(\tilde{z}(0), Y) \tag{11}$$

As shown in Equation 11, the total importance across all dimensions of $z$ corresponds to the decrease in $I(\tilde{z}(\lambda), Y)$ as the bottleneck narrows from fully open to fully closed, thereby satisfying the Sensitivity Axioms.

# F   PROOF OF IMPLEMENTATION INVARIANCE AXIOMS

By applying the chain rule, the proposed method inherently satisfies the Implementation Invariance Axiom.

# G   DETAILS OF $\lambda$

Although $\lambda$ remains consistent across dimensions within a layer, the actual updated values vary depending on the magnitude of each feature. For example, if a feature has a magnitude of 8 and $\lambda$ is set to (1/4), the final updated value will be 2. In contrast, if another feature dimension has a magnitude of 6, the updated value will be 1.5. These variations ensure that our theoretical properties hold and that the bottleneck effect is applied dynamically based on the specific characteristics of each feature.

# H   SENSITIVITY OF M2IB TO THE $\beta$ HYPERPARAMETER

Table 5: Effect of the $\beta$ Hyperparameter on Confidence Metrics for the M2IB Method Across Different Datasets

| Dataset | Conceptual Captions | | | | Imagenet | | | | Flickr8k | | | |
|---|---|---|---|---|---|---|---|---|---|---|---|---|
| $\beta$ | Img Conf Drop ↓ | Img Conf Incr ↑ | Text Conf Drop ↓ | Text Conf Incr ↑ | Img Conf Drop ↓ | Img Conf Incr ↑ | Text Conf Drop ↓ | Text Conf Incr ↑ | Img Conf Drop ↓ | Img Conf Incr ↑ | Text Conf Drop ↓ | Text Conf Incr ↑ |
| 0.01 | 0.8738 | 38.65 | 0.9779 | 44 | 0.835 | 52.6 | 1.1897 | 41.7 | 1.2544 | 27.3 | 1.1789 | 47.6 |
| 0.02 | 0.8886 | 39.05 | 0.93 | 45.25 | 0.8714 | 52.8 | 1.3856 | 35.8 | 1.2635 | 27.6 | 1.0784 | 48.8 |
| 0.03 | 0.9144 | 39.2 | 0.9591 | 46.35 | 0.9138 | 51.8 | 1.5526 | 32.6 | 1.282 | 28 | 1.174 | 47.6 |
| 0.04 | 0.943 | 39.15 | 1.077 | 45.4 | 0.9534 | 51.5 | 1.7488 | 30.8 | 1.3025 | 28.6 | 1.2631 | 47.7 |
| 0.05 | 0.9699 | 38.95 | 1.1631 | 44.7 | 0.9929 | 50.8 | 1.8996 | 31.4 | 1.3266 | 29 | 1.3423 | 46.4 |
| 0.06 | 1.0003 | 38.35 | 1.2813 | 42.9 | 1.0387 | 50 | 1.9928 | 29.9 | 1.3525 | 28.6 | 1.444 | 45.2 |
| 0.07 | 1.0325 | 38.15 | 1.3853 | 41.6 | 1.0842 | 50 | 2.1209 | 30.4 | 1.382 | 28 | 1.5843 | 43.3 |
| 0.08 | 1.0627 | 38.15 | 1.4937 | 39.75 | 1.1249 | 50 | 2.2748 | 27.1 | 1.4126 | 28.4 | 1.7559 | 41 |
| 0.09 | 1.0918 | 38.3 | 1.6044 | 39.15 | 1.1634 | 50.4 | 2.3936 | 25.9 | 1.444 | 28.5 | 1.8962 | 37.9 |
| 0.1 | 1.1244 | 38.4 | 1.7059 | 37.4 | 1.203 | 49.8 | 2.5389 | 24.7 | 1.4731 | 28.1 | 2.0783 | 34.7 |
| 0.2 | 1.4748 | 35.85 | 2.4205 | 27.85 | 1.4989 | 45.9 | 3.621 | 18.3 | 1.8176 | 27 | 2.8446 | 26.4 |
| 0.3 | 1.8328 | 32.1 | 2.7202 | 24.95 | 1.7226 | 42.4 | 4.1689 | 13.7 | 2.2708 | 24.7 | 3.0958 | 23.1 |
| 0.4 | 2.1324 | 29.95 | 2.8379 | 23.5 | 1.8277 | 40.8 | 4.3674 | 12.9 | 2.6872 | 22.5 | 3.2358 | 21.4 |
| 0.5 | 2.3452 | 28.4 | 2.9112 | 22.8 | 1.9042 | 39.9 | 4.4614 | 12.4 | 3.0238 | 19.9 | 3.2889 | 21.7 |
| 0.6 | 2.5184 | 25.95 | 2.9522 | 22.6 | 1.9561 | 38.8 | 4.5032 | 12.2 | 3.2989 | 18.2 | 3.3116 | 21.6 |
| 0.7 | 2.6748 | 25.3 | 2.9737 | 22.4 | 2.0037 | 37.5 | 4.5393 | 12.1 | 3.549 | 16.4 | 3.3198 | 21.7 |
| 0.8 | 2.8316 | 23.9 | 2.9863 | 22.35 | 2.0717 | 36.7 | 4.5558 | 12.1 | 3.7771 | 15.5 | 3.3254 | 21.7 |
| 0.9 | 2.9715 | 23.25 | 2.9947 | 22.15 | 2.1504 | 35.5 | 4.5649 | 12.1 | 3.9766 | 14.3 | 3.3289 | 21.7 |

As shown in the table 5, the $\beta$ hyperparameter has a significant impact on the performance of the M2IB method, indicating that the method is sensitive to variations in $\beta$. As $\beta$ increases, the values of both Image Confidence Drop (Img Conf Drop ↓) and Text Confidence Drop (Text Conf Drop ↓) increase noticeably across the Conceptual Captions, Imagenet, and Flickr8k datasets. This demonstrates that as $\beta$ grows, the model's performance deteriorates on these metrics. Similarly, the values of Image Confidence Increase (Img Conf Incr ↑) and Text Confidence Increase (Text Conf Incr ↑) decrease as $\beta$ increases, further illustrating the influence of this hyperparameter on model behavior.

For instance, when $\beta$ increases from 0.01 to 0.9, on the Conceptual Captions dataset, the Image Confidence Drop rises from 0.8738 to 2.9715, and the Text Confidence Drop rises from 0.9779 to 2.9947, while the Image Confidence Increase decreases from 38.65 to 23.25, and the Text Confidence Increase decreases from 44 to 22.15. This trend is consistent across other datasets, particularly when $\beta$ is larger, leading to more pronounced performance degradation. Therefore, it can be concluded that the M2IB method is highly sensitive to the $\beta$ hyperparameter, and tuning $\beta$ has a substantial effect on the confidence metrics of the model.

Table 6: Expanded the scope of num_steps ablation study

| Dataset | num_steps | target_layer | Img Conf Drop | Img Conf Incr | Text Conf Drop | Text Conf Incr |
|---|---|---|---|---|---|---|
| Conceptual Captions | 3 | 9 | 0.9649 | 42.5 | 0.2066 | 43.5 |
| | 8 | 9 | 0.9424 | 43.2 | 0.2409 | 43.2 |
| | 13 | 9 | 0.9422 | 42.8 | 0.3268 | 44.75 |
| | 18 | 9 | 0.9408 | 42.9 | 0.4288 | 45 |
| ImageNet | 3 | 9 | 0.9698 | 51.7 | 0.3746 | 56.3 |
| | 8 | 9 | 0.9438 | 53 | 0.4046 | 56.9 |
| | 13 | 9 | 0.9506 | 53 | 0.4444 | 57.3 |
| | 18 | 9 | 0.9648 | 53.9 | 0.4935 | 56 |
| Flickr8k | 3 | 9 | 1.4636 | 25.5 | 0.3875 | 51 |
| | 8 | 9 | 1.4526 | 26 | 0.4324 | 55 |
| | 13 | 9 | 1.4437 | 26.5 | 0.5525 | 53.6 |
| | 18 | 9 | 1.4463 | 27.1 | 0.7381 | 53 |

Table 7: Expanded the scope of target_layer ablation study

| Dataset | num_steps | target_layer | Img Conf Drop | Img Conf Incr | Text Conf Drop | Text Conf Incr |
|---|---|---|---|---|---|---|
| Conceptual Captions | 10 | 2 | 0.8577 | 41.95 | 1.1717 | 40 |
| | 10 | 4 | 0.8338 | 43.6 | 1.2319 | 37.2 |
| | 10 | 5 | 0.8106 | 43.95 | 1.5805 | 34.5 |
| | 10 | 6 | 0.8514 | 43.55 | 0.9867 | 40.1 |
| | 10 | 7 | 0.8911 | 43.75 | 0.8798 | 39.4 |
| | 10 | 8 | 0.8898 | 41.6 | 0.3655 | 43.75 |
| ImageNet | 10 | 2 | 0.7062 | 54.5 | 2.1586 | 33.7 |
| | 10 | 4 | 0.72 | 55.1 | 2.625 | 31.8 |
| | 10 | 5 | 0.7906 | 55 | 3.496 | 19.9 |
| | 10 | 6 | 0.7793 | 56.1 | 2.4207 | 32.5 |
| | 10 | 7 | 0.8931 | 54.8 | 1.4745 | 47.4 |
| | 10 | 8 | 0.9258 | 52.7 | 0.985 | 49.8 |
| Flickr8k | 10 | 2 | 1.2641 | 28.1 | 1.2748 | 44.4 |
| | 10 | 4 | 1.283 | 27.7 | 1.4821 | 40.9 |
| | 10 | 5 | 1.247 | 27.4 | 2.0031 | 36.5 |
| | 10 | 6 | 1.2875 | 28.3 | 1.1981 | 46.9 |
| | 10 | 7 | 1.3609 | 28.6 | 1.2775 | 43.1 |
| | 10 | 8 | 1.2881 | 28.8 | 0.9316 | 46.3 |

# I  EXPANDED ABLATION STUDY OF *num_steps* AND *target_layer*

We expanded the scope of our ablation studies with additional results, as shown in Tables 6 and 7 in the supplementary material, to provide a more comprehensive analysis of hyperparameter interactions.

Table 8: Forward and backward passes of NIB compared to other methods

| Method | Forward | Backward |
|---|---|---|
| NIB | 12 | 10 |
| RISE | 301 | 0 |
| Grad-CAM | 3 | 2 |
| Chefer et al. | 3 | 0 |
| SM | 3 | 0 |
| MFABA | 21 | 10 |
| M2IB | 22 | 20 |
| FastIG | 3 | 0 |

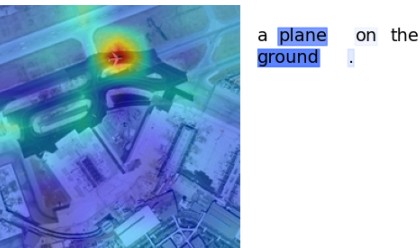

Figure 2: Attribution result of RSICD dataset

## J   COMPUTATIONAL EFFICIENCY

As outlined in Table 8, NIB achieves superior efficiency in terms of forward and backward passes compared to other methods. The efficiency of our method scales independently of model complexity and dataset size, as demonstrated in our evaluation.

## K   ATTRIBUTION RESULTS

To provide further insights into generalization, we have generated attribution examples using the RSICD remote sensing dataset. Figure 2 indicates that the proposed method could generalise effectively to other domains.

