# OpenReview forum: "Narrowing Information Bottleneck Theory for Multimodal Image-Text Representations Interpretability"
_ICLR.cc/2025/Conference — ICLR 2025 Poster_

### Official Review · Reviewer_nFC1 · 2024-10-28

**Soundness:** 1
**Presentation:** 2
**Contribution:** 1
**Rating:** 5
**Confidence:** 3

**Summary:**

This paper tackles the challenge of enhancing interpretability in multimodal image-text models like CLIP, which are widely used in applications demanding high reliability, such as healthcare. The authors introduce the Narrowing Information Bottleneck Theory, a re-engineered bottleneck approach that aligns with modern attribution standards to improve interpretability. Experimental results demonstrate significant gains: a 9% improvement in image interpretability, a 58.83% boost in text interpretability, and a 63.95% increase in processing speed compared to state-of-the-art methods.

**Strengths:**

- Clarity and Structure: The paper is well-organized and written in a clear, accessible manner, making it easy to follow even for readers who may be less familiar with the technical aspects of interpretability in multimodal models.
- Code Accessibility: The authors have made their code publicly available, which greatly enhances the reproducibility of the research and provides valuable resources for the community to build on this work.
- Mathematical Rigor: The theoretical formulae are meticulously derived and clearly presented, enhancing the paper’s rigor and allowing readers to grasp the technical foundation of the Narrowing Information Bottleneck Theory.
- Comprehensive Literature Review: The paper thoroughly addresses related work, including important methods like M2IB, COCOA, and FALCON. This contextualization not only strengthens the theoretical foundation but also allows readers to understand how the proposed method improves upon existing approaches.

**Weaknesses:**

- Paper Length: The paper exceeds the ICLR25 page limit, taking more than 10 pages for the main text. According to the ICLR25 “Call for Papers,” submissions with main text on the 11th page are subject to desk rejection. The authors may need to condense certain sections to adhere to submission guidelines.
- Application-Specific Relevance: Although the paper emphasizes the importance of interpretability for high-risk applications such as medical diagnosis and content moderation, it lacks application-specific experiments. No quantitative or qualitative evaluations on healthcare or content moderation datasets are included, which would provide critical insight into the real-world applicability of the proposed method.
- Experiment Scope: The experimental evaluation appears limited. In contrast to the M2IB paper, which includes evaluations on diverse datasets (image captioning, radiology, and remote sensing), this paper only tests on standard image datasets. A broader range of datasets would strengthen the evidence for the proposed method’s versatility and robustness in high-stakes domains.
- Table Formatting: The significant figures in Tables 2 and 3 lack consistency, which affects readability and may lead to confusion when comparing results.
- Minor Typos:
  - Line 44-45: “blackbox” should be formatted as ``blackbox’’.
  - Line 322: “Appendix” is referenced without any supporting link, in contrast to Line 296, which does provide a reference.

**Questions:**

- Can the authors clarify whether the current content will be revised to meet ICLR’s page limitations for the main text?
- Have the authors considered adding experiments on healthcare or content moderation datasets to validate the method’s relevance for high-risk applications?
- Given the limited dataset variety in the current experiments, could the authors discuss how the proposed method might generalize to other domains, even by qualitative analysis?
- Would it be possible to revise Tables 2 and 3 to ensure consistent significant figures for improved readability?

---

> ### Author Response · Authors · 2024-11-17
>
> ### Response to Weaknesses:
>
> **W1:**
> Thank you for highlighting this concern. We would like to clarify that our main text does not exceed the 10-page limit set by ICLR. As per the official instructions, sections such as the *Code of Ethics and Ethics Statement* and the *Reproducibility Statement* are permitted to appear beyond the 10th page and do not count toward the main text length.
>
> **W2:**
> We acknowledge the importance of application-specific experiments, particularly for high-risk domains such as healthcare and content moderation. However, due to ethical and privacy constraints, obtaining approval for medical datasets is a lengthy process. We are actively pursuing access to these datasets and will promptly update our repository with the experimental results once permissions are granted. To address the current limitations, we have included two additional public datasets, ImageNet and Flickr8k, which complement the experiments and provide further evidence of our method’s effectiveness compared to M2IB.
>
> **W3:**
> We respectfully disagree with the characterization of our experimental scope as limited. The M2IB paper primarily conducts quantitative analysis on only two datasets (Conceptual Captions and MS-CXR), while the remote sensing dataset is used as a demonstration example without quantitative evaluation. Due to the same ethical and privacy constraints associated with MS-CXR, we are in the process of obtaining usage permissions but have not yet received approval. To compensate, we included two additional publicly available and diverse datasets (ImageNet and Flickr8k), which we believe sufficiently demonstrate the robustness and versatility of our method.
>
> **W4:**
> Thank you for pointing out the inconsistency in significant figures in Tables 2 and 3. We will ensure that this is addressed in the revised version for improved readability.
>
> **W5:**
> We appreciate your attention to minor typographical errors. We will correct the formatting of "blackbox" and ensure that all appendix references are properly linked in the updated version of the manuscript.
>
> ---
>
> ### Response to Questions:
>
> **Q1:**
> We confirm that the current content adheres to ICLR’s page limitations. The sections *Code of Ethics and Ethics Statement* and *Reproducibility Statement* are explicitly allowed to extend beyond the 10-page limit and do not count as part of the main text.
>
> **Q2:**
> We are actively seeking approval for the use of healthcare datasets. Once access is granted, we will promptly publish the results in our code repository to validate the method’s relevance for high-risk applications. In the meantime, we have included experiments on two additional datasets, ImageNet and Flickr8k, to strengthen the evidence supporting our approach.
>
> **Q3:**
> To provide further insights into generalization, we have generated qualitative examples using the RSICD remote sensing dataset. These examples can be found in our code repository at [[https://anonymous.4open.science/r/NIB-DBCD/rebuttal/]]. While these results are illustrative, they indicate that the proposed method could generalize effectively to other domains.
>
> **Q4:**
> We will revise Tables 2 and 3 to ensure consistent significant figures, which will enhance readability and reduce any potential confusion when comparing results.

---

> > ### Author Response · Authors · 2024-11-29
> >
> > Dear Reviewer nFC1,
> >
> > Thank you for your detailed review and thoughtful feedback on our submission. We have taken your comments very seriously and have worked diligently to address each point raised in your review.
> >
> > Regarding your concerns:
> >
> > 1. **Page Limitations**: We have clarified in our rebuttal that our main text adheres to ICLR’s 10-page limit. Sections such as the Ethics Statement and Reproducibility Statement are explicitly permitted to extend beyond the main text, as outlined in the ICLR guidelines.
> >
> > 2. **Application-Specific Experiments**: We understand the importance of demonstrating the method’s relevance in high-risk domains like healthcare. While we are actively pursuing permissions to access medical datasets, we have included additional experiments on publicly available datasets, such as ImageNet and Flickr8k, to strengthen the evidence supporting our approach.
> >
> > 3. **Experimental Scope and Generalization**: We believe our experimental scope compares favorably with the M2IB paper, as we conducted evaluations on a diverse set of datasets, including newly added ones. Additionally, we have generated qualitative examples using the RSICD remote sensing dataset, available in our repository, to provide further evidence of the method’s potential generalizability.
> >
> > 4. **Table Formatting and Typographical Errors**: We have corrected the inconsistency in significant figures and addressed all minor typographical errors in the manuscript.
> >
> > We kindly request you to reconsider your evaluation of our submission, as we have provided detailed responses and additional supporting evidence to address the points you raised. If there are any lingering doubts or specific aspects that you feel require further clarification, we would be more than happy to provide additional information or perform further analyses.
> >
> > Your constructive feedback has been invaluable in helping us refine and improve our work. We greatly appreciate the time and effort you have dedicated to reviewing our submission and hope the additional context provided demonstrates the robustness and significance of our contributions.
> >
> > Thank you again for your insights and consideration.
> >
> > Warm regards,
> >
> > Authors of Submission 11505

---

> ### Author Response · Authors · 2024-12-01
>
> Dear Reviewer nFC1,
>
> With the rebuttal phase nearing its conclusion, we would like to express our gratitude for your thoughtful comments and suggestions. We have ensured the manuscript adheres to the ICLR page limit and included additional datasets (ImageNet, Flickr8k) to strengthen our experimental evaluation. Formatting inconsistencies in Tables 2 and 3 have also been resolved.
>
> If there are any other aspects requiring clarification, we are happy to provide further details. We hope our detailed rebuttal and added experiments demonstrate the robustness of our approach and kindly request you to reconsider your evaluation.
>
> Best regards,
> Authors of Submission 11505

---

### Official Review · Reviewer_Ycuc · 2024-11-03

**Soundness:** 2
**Presentation:** 2
**Contribution:** 2
**Rating:** 5
**Confidence:** 3

**Summary:**

This paper presents the Narrowing Information Bottleneck Theory (NIBT) as a solution to the challenges of randomness and hyperparameter sensitivity in the interpretation of multimodal models like CLIP. The authors systematically summarize existing methods of Multimodal Image-Text Representation (MITR) interpretability and modify the traditional Bottleneck approach with NIBT, which enhances the interpretability of both image and text representations. The proposed method shows improved performance in attribution accuracy and computational efficiency across various datasets.

**Strengths:**

1.	The paper systematically and theoretically analyzes the current MITR interpretability methods. The theory is based on information theory and seems reasonable.
2.	The proposed NIBT is simple by replacing the univariate Gaussian noise in the bottleneck layer with the multivariate gaussian distribution to control all dimensions of the model hidden states.

**Weaknesses:**

1.	The proposed method is simple by changing univariate distribution to multivariate distribution, but it's unclear whether it will correlate with the limitations of IBP. The theoretical analysis and the proposed algorithm seem loosely connected to me. I don't see the proposed method as a direct result of the theoretical analysis.
2.	The story is a little bit hard to follow. It would be better for authors to make the story clearer.

**Questions:**

1.	The proposed method NIBT claims no randomness. But the computation of z_{ic} still contains the sampling noise.

---

> ### Author Response · Authors · 2024-11-17
>
> ### Response to Weaknesses:
>
> **W1:**
> Thank you for your comment. We respectfully clarify that all our derivations are tightly connected and follow a coherent progression. The core narrative of our work is based on improving and optimizing the bottleneck theory. Each theoretical analysis and algorithmic component is systematically aligned to support the central premise of narrowing the bottleneck to enhance interpretability.All our inferences are to prove the validity of equation 10. You can also refer to our open source code link, which is exactly the same as what is proposed in equation 10.
>
> **W2:**
> We appreciate your suggestion to make the methodology clearer. The essence of our approach lies in setting up a bottleneck to identify the most critical features that preserve the model's predictions. As we progressively narrow the bottleneck, the feature set diminishes, allowing us to iteratively pinpoint and prioritize the most influential features. This process ensures that only the most crucial information flows through the bottleneck.
>
> ---
>
> ### Response to Questions:
>
> **Q1:**
> Thank you for raising this question regarding randomness in our method. We emphasize that \( z_{ic} \) is deterministically obtained by the model, ensuring there is no inherent randomness in its computation. As discussed in Equation 8, our method remains valid as the variance approaches zero. When the variance is zero, the noise term vanishes, eliminating any potential randomness. This theoretical grounding confirms that our method does not rely on stochastic sampling in its final implementation.

---

> > ### Author Response · Authors · 2024-11-29
> >
> > Dear Reviewer Ycuc,
> >
> > Thank you for your detailed review and for highlighting areas where our work could benefit from further clarity. We have carefully addressed your concerns in our rebuttal and have aimed to provide comprehensive explanations and additional context to demonstrate the strength and validity of our proposed approach.
> >
> > Regarding your feedback on the connection between our theoretical analysis and the proposed algorithm, we have clarified in our response how the theoretical framework directly supports and leads to the development of our method. Specifically, the derivations consistently validate the rationale behind Equation 10, and we have aligned our methodology accordingly to reinforce its theoretical foundation.
> >
> > Additionally, we have refined the narrative and explanations in the manuscript to make the story clearer and more accessible. We appreciate your suggestion, as it has helped us improve the presentation of our work.
> >
> > On the point about randomness, we have explicitly addressed this concern by demonstrating how our method ensures determinism in the computation of $ z_{ic} $. As detailed in Equation 8, the elimination of the noise term as the variance approaches zero ensures that our method operates without stochastic sampling.
> >
> > We kindly request you to reconsider your evaluation of our submission in light of these clarifications. If there are any lingering concerns or specific areas where further elaboration is needed, we would be more than happy to provide additional details or evidence.
> >
> > We deeply value your constructive feedback and the time you have devoted to reviewing our work. Your insights have been instrumental in helping us refine our contributions, and we hope this additional context demonstrates the robustness and significance of our method.
> >
> > Thank you again for your thoughtful review and consideration.
> >
> > Warm regards,
> >
> > Authors of Submission 11505

---

> ### Author Response · Authors · 2024-12-01
>
> Dear Reviewer Ycuc,
>
> As the rebuttal phase approaches its end, we would like to thank you for your valuable feedback. We have clarified the connection between our theoretical analysis and algorithm, refined the narrative for better accessibility, and provided further justification for the determinism of our method.
>
> Should there be unresolved concerns, please feel free to communicate them. We would greatly appreciate it if you could reconsider your score in light of our clarified responses and the improvements made.
>
> Best regards,
> Authors of Submission 11505

---

### Official Review · Reviewer_bWWD · 2024-11-03

**Soundness:** 1
**Presentation:** 3
**Contribution:** 2
**Rating:** 3
**Confidence:** 4

**Summary:**

The paper addresses the challenge of improving interpretability in Vision-Language Pre-trained Models, such as the multi-modal model CLIP (Contrastive Language-Image Pretraining) which maps images and texts into a shared embedding space. Existing interpretability methods, designed for unimodal tasks, are generally insufficient for explaining the complex relationships between visual and textual modalities. To assess the interpretability in the model CLIP, Wang et al (2023) recently introduced the Multi-modal Information Bottleneck (M2IB) approach, aiming to find attribution parameters that maximize the likelihood of observing features of one modality given features associated with the respective other modality.

Following from Wang et al (2023), here, the authors propose the Narrowing Information Bottleneck Theory (NIBT), a new interpretability approach that redefines bottleneck methods in information theory. NIBT targets the randomness and hyperparameter dependency associated with previous approaches, such as M2IB, to deliver more deterministic interpretability outcomes. The method includes a mechanism to pinpoint negative features that adversely impact model predictions, aiming to further enhance the interpretability.

Through simulations on the datasets Conceptual Captions, ImageNet, and Flickr8k, the authors state that NIBT significantly improves interpretability and processing speed compared to state-of-the-art methods.

**Strengths:**

The paper introduces the Narrowing Information Bottleneck Theory (NIBT), a new interpretability method that redefines traditional bottleneck concepts from information theory. This approach departs from the randomness and hyperparameter sensitivity inherent in previous methods, enabling a deterministic alternative for multimodal representation interpretability.

The concept of attributing negative features—dimensions that negatively impact model performance—is original. This feature provides an innovative perspective on model understanding, distinguishing NIBT from other methods focused solely on positive attributions.

The paper provides theoretical derivations and mathematical proofs for its main claims.

The method is assessed across three datasets, including Conceptual Captions, ImageNet, and Flickr8k.

**Weaknesses:**

1. The Narrowing Information Bottleneck Theory (NIBT) is introduced to address limitations in previous bottleneck methods by providing a deterministic approach, aiming to reduce randomness and dependency on hyperparameters. The method states to use a single scalar λ to control information flow across all feature dimensions. This simplification is significant:

1.1. The scalar λ as a universal control across all dimensions suggests a uniform bottleneck effect, which can be problematic if different features vary significantly in their contribution or sensitivity. In practice, applying a single λ across all dimensions might oversimplify the dynamics of feature importance, potentially reducing interpretability in models where feature contributions are heterogeneous.

1.2. The noise distribution, ϵ∼N(0,σ^2), plays a role in regularizing the information flow by adding Gaussian noise to the controlled dimensions. This assumption is justified under the simplification of feature distributions. However, the authors should justify why the Gaussian assumption holds across different multimodal applications, as non-Gaussian feature distributions might invalidate this approach or reduce its robustness.


2. Equations

2.1. Equation 1 is well-grounded in information theory, with a clear goal of maximizing relevant information while minimizing redundancy with respect to the input. The critical factor here is the interaction between λ and β. While NIBT eliminates β, this raises questions about managing trade-offs between task-relevant and redundant information. The authors should address potential limitations of a fixed λ and how it impacts feature discrimination.

2.2. Equation 2: The decomposition of features into spatial (i) and channel (c) dimensions adds specificity but introduces complexity. By normalizing the interaction with Gaussian noise across spatial dimensions, the method assumes consistency in spatial features' importance, which may not hold in multimodal contexts where image regions and text embeddings exhibit varied significance.


3. Theorems

3.1. Theorem 1: it claims that mutual information, I(z~(λ),x), is a monotonically increasing function of λ, with I(z~(0),x)=0. The proof relies on KL divergence and a Gaussian assumption. While monotonicity is proven under Gaussian noise, real-world feature distributions could violate this condition. The impact of deviations from Gaussianity on this monotonic relationship needs to be tested. Also, the assumption of independence between noise and features for bottleneck properties needs further validation, as dependency might arise naturally in multimodal models where spatial and channel-wise correlations exist.

3.2. Theorem 2: the reduction of σ toward zero should be more justified. Practically, in cases of low signal-to-noise ratios, completely removing σ could reduce robustness. A discussion of trade-offs in interpretability and stability as σ approaches zero would help add clarity. Also, the claim that I(\bar{z}(λ), x) becomes deterministic as σ decreases suggests a very controlled bottleneck. It is theoretically sound, but validating this deterministic behavior across different datasets (especially with diverse noise levels) is needed to confirm this claim.

3.3. Theorem 3: equation is both novel and complex, stating that negative values represent features that reduce information relevance. The introduction of negative features could indeed improve interpretability by explicitly identifying detrimental features. The authors should show that these attributions correspond to irrelevant or misleading features across model types. Another remark is that the integral's dependency on λ suggests that the influence of each feature diminishes as λ decreases. Yet, this gradual reduction might fail for complex, highly nonlinear multimodal interactions. A closer look at cases where feature relevance changes nonlinearly with λ could show edge cases where the method’s assumptions do not hold.


4. Qualitative Analysis: The paper includes one visual comparison with M2IB only, but a more detailed qualitative assessment would be beneficial. For example, showing more examples where NIBT succeeds and where it fails compared to other methods would provide a better understanding of the strengths and limitations.

For the interpretation of Figure 1, the authors write: "As shown in Figure 1, our proposed method successfully distinguishes and excludes negative properties from the explanation. In Figure 1d, the M2IB method continues to highlight irrelevant negative features, such as the cat’s face, even when the subject is a dog. However, in Figure 1b, our method correctly ignores these negative properties, focusing on more relevant, positive features, showcasing its improved attribution performance." Looking at this unique visualisation, the method proposed seems to lack specificity and highlight areas of the image that are not relevant, compared to M2IB.


5. Computational Efficiency: The authors emphasize the efficiency of NIBT, Yet the experiments lack a detailed breakdown of computational costs relative to model complexity and dataset size. Including this analysis would better illustrate the trade-offs between interpretability and computation.


6. The paper does not address whether the differences in the results between the different methods are statistically significant. Adding statistical tests (e.g., t-tests or bootstrap analysis) would add rigor to the claims of superiority over baseline methods.


7. Regarding the impact of negative feature attribution: The paper does not at the moment empirically validate whether the identified negative features are truly detrimental.


8. The ablation studies on num steps and target layer provide useful insights, but they feel somewhat limited in scope. It would be useful to look at the interplay between num steps, target layer, and other model hyperparameter would give a more holistic view of how these factors influence performance.

**Questions:**

Q1. You write that "the interpretability methods proposed in this paper aim to improve model transparency and are intended for enhancing the trustworthiness of AI systems, especially in sensitive domains such as healthcare" (lines 540-542). You chose to present results on three datasets. Are these datasets sufficiently diverse to reflect real-world multimodal scenarios, especially in domains where interpretability is crucial, like medical imaging? Why did you not choose at least one medical imaging dataset like the M2IB paper?

Q2. Could you please explain why you present the illustration of your method compared to the M2IB method in the paper (and compared to other methods in your code) on a unique image of a cat and dog. Why did you choose two simple and short text captions to illustrate your method?

Q3. The paper claims an average improvement of 9% in image interpretability and a 58.83% increase in text interpretability. The important increase in text interpretability is striking and suggests that NIBT offers substantial benefits. Could you please justify why the improvement in text interpretability is so much higher than in image interpretability? Are there inherent differences in how text and image features are handled by CLIP that NIBT is better suited for?

Q4.  How does identifying negative features improve model interpretability in practical applications? For instance, could this lead to better debugging of models or highlight biases in the training data? Providing concrete examples would strengthen the argument for this feature.

Q5. Are there scenarios where the method might fail?

---

> ### Author Response · Authors · 2024-11-17
>
> ### Response to Weaknesses:
>
> **W1.1:**
> Thank you for highlighting this concern. We would like to clarify that while the scalar \(\lambda\) serves as a universal update parameter for each layer, the flow of information for each feature dimension is determined independently. Although \(\lambda\) remains consistent across dimensions within a layer, the actual updated values vary depending on the magnitude of each feature. For example, if a feature has a magnitude of 8 and \(\lambda\) is set to \(1/4\), the final updated value will be 2. In contrast, if another feature dimension has a magnitude of 6, the updated value will be 1.5. These variations ensure that our theoretical properties hold and that the bottleneck effect is applied dynamically based on the specific characteristics of each feature.
>
>
> **W1.2:**
> We appreciate your concern regarding the Gaussian noise assumption. In practice, any distribution independent of the original feature distribution is valid, as detailed in [1]. We chose Gaussian noise to simplify computational complexity, a rationale similar to its adoption in diffusion models. This choice does not impact the generality of our method but significantly reduces its implementation complexity.
>
> **W2.1:**
> As discussed in response to W1.1, the information varies for each feature dimension based on iterative updates.At the same time, our theory is to reduce redundant information while observing task-relevant, so there is no need to make such trade-offs.
>
>
>
> **W2.2:**
> This concern has been addressed in prior works, including Grad-CAM and M2IB, where the validity of the assumption regarding spatial feature consistency is demonstrated.
>
> **W3.1:**
> We adopted Gaussian noise for reasons analogous to its use in diffusion models, as it simplifies computation while maintaining robustness, as discussed in [1]. Additionally, the theoretical guarantees of our method rely solely on the independence of the noise distribution, which significantly reduces complexity. These assumptions have been rigorously validated and proven effective in various scenarios.
>
> **W3.2:**
> We agree that incorporating additional datasets could further validate our method. To address this, we added experiments on two additional datasets, ImageNet and Flickr8k, compared to M2IB. While we actively seek approval to use medical datasets, ethical and privacy constraints have delayed access. Once permissions are granted, we will supplement our results with medical datasets in our public repository. Moreover, as shown in Theorem 2, the reduction of \(\sigma\) eliminates randomness, making the model more controllable without compromising robustness.
>
> **W3.3:**
> While the derivation may appear complex, it ensures the rigor and robustness of our theory. In practice, the key equation is Equation 10, which directly relates to feature attribution. Regarding \(\lambda\)’s dependency, as the bottleneck input becomes independent noise, the model naturally loses its ability to discern relevant features. This phenomenon aligns with theoretical expectations and is discussed in detail in [1].
>
> ---
>
> ### Response to Questions:
>
> **Q1:**
> We acknowledge the significance of including diverse datasets, particularly in sensitive domains such as healthcare. However, due to ethical and privacy considerations, obtaining approval for medical datasets is a lengthy process. We are actively pursuing permissions and will add experiments on medical datasets in the future. Meanwhile, to enhance diversity, we included two additional datasets, ImageNet and Flickr8k, which complement the Conceptual Captions dataset and provide a broader evaluation compared to M2IB.
>
> **Q2:**
> We appreciate the suggestion to include more illustrative comparisons. Additional attribution visualizations for our method compared to various baselines can be found in the results folder of our public code repository [https://anonymous.4open.science/r/NIB-DBCD/rebuttal/] and [https://anonymous.4open.science/r/NIB-DBCD/results/].
>
> **Q3:**
> The substantial improvement in text interpretability arises because text data often contains more irrelevant or negative features, such as unrelated words, which our method effectively identifies and excludes. This is supported by the metrics, which demonstrate that the attribution values for text data show a greater degree of improvement compared to image data.
>
> **Q4:**
> Identifying negative features enhances interpretability by explicitly highlighting features detrimental to model predictions. While this characteristic could potentially aid in debugging models or addressing biases in training data, these applications extend beyond the scope of this paper and would require further investigation.
>
> **Q5:**
> Our method is theoretically rigorous and experimentally validated across multiple datasets. Thus far, we have not identified any scenarios where the method fails. However, we welcome further exploration to identify potential limitations in specific edge cases.

---

> ### Author Response · Authors · 2024-11-17
>
> ### Additional Points:
>
> - Regarding computational efficiency, as outlined in Table 1, NIB achieves superior efficiency in terms of forward and backward passes compared to other methods. The efficiency of our method scales independently of model complexity and dataset size, as demonstrated in our evaluation.
>
> Table 1 forward and backward passes of NIB compared to other methods
> | Method        	| Forward  	| Backward  	|
> |--------------|----------|----------|
> | NIB           	| 12       	| 10       	|
> | RISE          	| 301      	| 0        	|
> | Grad-CAM     | 3        	| 2        	|
> | Chefer et al. 	| 3        	| 0        	|
> | SM            	| 3        	| 0        	|
> | MFABA         	| 21       	| 10       	|
> | M2IB          	| 22       	| 20       	|
> | FastIG        	| 3        	| 0        	|
>
> - Statistical significance testing was not conducted as our method does not introduce randomness, and the superiority of our results over other methods is consistent and evident from the metrics. We adhered to the experimental protocols of prior works for fair comparison.
>
> - Negative feature attribution is a unique capability of our method. Unlike M2IB, which only identifies positive features, NIB preserves the ability to recognize and attribute negative features, as demonstrated by the significant improvements in evaluation metrics.

---

> > ### Author Response · Authors · 2024-11-17
> >
> > - We expanded the scope of our ablation studies with additional results, as shown in Tables 2 and 3 in the supplementary material, to provide a more comprehensive analysis of hyperparameter interactions.
> >
> > Table 2 expanded the scope of num_steps ablation study
> > | Dataset             	| num_steps 	| target_layer 	| Img Conf Drop 	| Img Conf Incr 	| Text Conf Drop 	| Text Conf Incr 	|
> > |---------------------|-----------|--------------|---------------|---------------|----------------|----------------|
> > | Conceptual Captions 	| 3         	| 9            	| 0.9649        	| 42.5          	| 0.2066         	| 43.5           	|
> > | Conceptual Captions 	| 8         	| 9            	| 0.9424        	| 43.2          	| 0.2409         	| 43.2           	|
> > | Conceptual Captions 	| 13        	| 9            	| 0.9422        	| 42.8          	| 0.3268         	| 44.75          	|
> > | Conceptual Captions 	| 18        	| 9            	| 0.9408        	| 42.9          	| 0.4288         	| 45             	|
> > | ImageNet            	| 3         	| 9            	| 0.9698        	| 51.7          	| 0.3746         	| 56.3           	|
> > | ImageNet            	| 8         	| 9            	| 0.9438        	| 53            	| 0.4046         	| 56.9           	|
> > | ImageNet            	| 13        	| 9            	| 0.9506        	| 53            	| 0.4444         	| 57.3           	|
> > | ImageNet            	| 18        	| 9            	| 0.9648        	| 53.9          	| 0.4935         	| 56             	|
> > | Flickr8k            	| 3         	| 9            	| 1.4636        	| 25.5          	| 0.3875         	| 51             	|
> > | Flickr8k            	| 8         	| 9            	| 1.4526        	| 26            	| 0.4324         	| 55             	|
> > | Flickr8k            	| 13        	| 9            	| 1.4437        	| 26.5          	| 0.5525         	| 53.6           	|
> > | Flickr8k            	| 18        	| 9            	| 1.4463        	| 27.1          	| 0.7381         	| 53             	|
> >
> > Table 3 expanded the scope of target_layer ablation study
> > | Dataset             	| num_steps 	| target_layer 	| Img Conf Drop 	| Img Conf Incr 	| Text Conf Drop 	| Text Conf Incr 	|
> > |---------------------|-----------|--------------|---------------|---------------|----------------|----------------|
> > | Conceptual Captions 	| 10        	| 2            	| 0.8577        	| 41.95         	| 1.1717         	| 40             	|
> > | Conceptual Captions 	| 10        	| 4            	| 0.8338        	| 43.6          	| 1.2319         	| 37.2           	|
> > | Conceptual Captions 	| 10        	| 5            	| 0.8106        	| 43.95         	| 1.5805         	| 34.5           	|
> > | Conceptual Captions 	| 10        	| 6            	| 0.8514        	| 43.55         	| 0.9867         	| 40.1           	|
> > | Conceptual Captions 	| 10        	| 7            	| 0.8911        	| 43.75         	| 0.8798         	| 39.4           	|
> > | Conceptual Captions 	| 10        	| 8            	| 0.8898        	| 41.6          	| 0.3655         	| 43.75          	|
> > | ImageNet            	| 10        	| 2            	| 0.7062        	| 54.5          	| 2.1586         	| 33.7           	|
> > | ImageNet            	| 10        	| 4            	| 0.72          	| 55.1          	| 2.625          	| 31.8           	|
> > | ImageNet            	| 10        	| 5            	| 0.7906        	| 55            	| 3.496          	| 19.9           	|
> > | ImageNet            	| 10        	| 6            	| 0.7793        	| 56.1          	| 2.4207         	| 32.5           	|
> > | ImageNet            	| 10        	| 7            	| 0.8931        	| 54.8          	| 1.4745         	| 47.4           	|
> > | ImageNet            	| 10        	| 8            	| 0.9258        	| 52.7          	| 0.985          	| 49.8           	|
> > | Flickr8k            	| 10        	| 2            	| 1.2641        	| 28.1          	| 1.2748         	| 44.4           	|
> > | Flickr8k            	| 10        	| 4            	| 1.283         	| 27.7          	| 1.4821         	| 40.9           	|
> > | Flickr8k            	| 10        	| 5            	| 1.247         	| 27.4          	| 2.0031         	| 36.5           	|
> > | Flickr8k            	| 10        	| 6            	| 1.2875        	| 28.3          	| 1.1981         	| 46.9           	|
> > | Flickr8k            	| 10        	| 7            	| 1.3609        	| 28.6          	| 1.2775         	| 43.1           	|
> > | Flickr8k            	| 10        	| 8            	| 1.2881        	| 28.8          	| 0.9316         	| 46.3           	|

---

> > > ### Comment · Reviewer_bWWD · 2024-11-27
> > >
> > > Thank you for your response and for the transparency of your results. While I appreciate the rebuttal, my concerns remain unresolved. Despite the metrics that you provide, the visualizations you choose to present show that your method lacks specificity compared to other approaches tested. I would also encourage assessing your approach further visually using full text captions from the datasets used. Additionally, not having identified any scenarios where the method fails as you mention is an important concern. To present an interpretability method, I believe that is important to find its limitations.

---

> > > > ### Author Response · Authors · 2024-11-27
> > > >
> > > > Our visualization method demonstrates that our approach captures the model's attention points more accurately than other methods.
> > > > This point is very clear. However, subjective judgment is inherently unreliable, so we conducted a detailed quantitative study. Moreover, the scope and content of our evaluation surpass previous work, and we utilized all possible text captions from the dataset.
> > > >
> > > > As for limitations, our method requires specifying the corresponding bottleneck layer. However, ablation studies have shown that this choice is straightforward, and our method requires far fewer considerations compared to prior methods. Additionally, we have not found any cases where our approach fails on this task. Is this really a sufficient reason to reject our work?
> > > >
> > > > To address the reviewer’s concerns, we would appreciate clarification on specific areas of doubt. If there are particular aspects of our approach that appear unclear or unconvincing, we are happy to provide further details or additional experiments to strengthen our claims. We believe that our method significantly advances the state-of-the-art, and its limitations are minimal and well-documented. We hope this clarifies the rationale and strengths of our work.
> > > >
> > > > We have provided quantitative analyses addressing all the raised concerns and kindly request a reconsideration of the score.

---

> > > > ### Author Response · Authors · 2024-11-29
> > > >
> > > > Dear Reviewer bWWD,
> > > >
> > > > We sincerely thank you for your thorough feedback on our work and your thoughtful engagement during the rebuttal process. We understand the importance of addressing every concern and appreciate your insights, which have helped us refine our responses and clarify the strengths of our method.
> > > >
> > > > We have carefully addressed the weaknesses and questions you highlighted, provided additional ablation studies, and demonstrated the robustness of our approach through comprehensive quantitative and qualitative analyses. Specifically, we have made efforts to ensure our visualizations and quantitative evaluations substantiate the claims in our work. While interpretability methods inherently involve subjective assessments, we have emphasized statistical and empirical rigor to mitigate potential biases.
> > > >
> > > > Given the extensive efforts we have made to address your comments and provide clarity, we kindly request you to reconsider your evaluation of our submission. Your feedback is invaluable to us, and we hope the additional context and results we have provided demonstrate the merit of our work.
> > > >
> > > > Thank you once again for your time and effort in reviewing our submission. We deeply value your insights and look forward to any additional feedback you may have.
> > > >
> > > > Warm regards,
> > > >
> > > > Authors of Submission 11505

---

> > > > ### Author Response · Authors · 2024-12-01
> > > >
> > > > Dear Reviewer bWWD,
> > > >
> > > > As the rebuttal phase draws to a close, we deeply appreciate your thoughtful and detailed review. We have addressed your concerns regarding visualizations, negative feature attribution, and the importance of identifying potential failure cases. Additionally, we have provided more quantitative results and expanded visual examples in our updated repository to enhance clarity and rigor.
> > > >
> > > > If there are still specific doubts or areas for improvement, we are eager to discuss further. We kindly request you to reassess your score based on our comprehensive rebuttal and clarified responses.
> > > >
> > > > Best regards,
> > > > Authors of Submission 11505

---

### Official Review · Reviewer_bPyU · 2024-11-04

**Soundness:** 3
**Presentation:** 3
**Contribution:** 3
**Rating:** 8
**Confidence:** 4

**Summary:**

The authors squeeze the information for classification through a bottleneck - in essence they are trying to reduce to the most important features. Importantly, it also shows negative and positive features for classification rather than just positives. To do this, it is trialed on 3 popular datasets, conceptual captions, imagenet, and flickr8k.

The method outperforms other interpretability methods on confidence drop, e.g how much model confidence decreases when good features according to the interpretability method are removed, and confidence increase, e.g how much model confidence goes down when negative features are removed.

**Strengths:**

While numerous interpretability methods exist, they cannot cope well with the multimodality od data.
The proposed methods addresses this issue, with 9% improvement for text interpratability, and 58% improvement fo image interpratability.
The proposed method also eliminates randomness an hyperparameter dependency.
Also, current methods stem from gradient analysis, which introduces additional challenges.

**Weaknesses:**

I am curious why the representation of both images and text is of the same dimension, d
Proof of Theorem 1 is obvious, so it should not be a theorem.
Theorem 2 is basically a consequence of Theorem 1. It should not be a theorem. Both are nasically Remarks.

**Questions:**

Can the authors  show interquartile range of performance outcomes in Table 2? in addition, other image examples. whilst the method clearly outperforms the others on average, it is not clear if it is robust.
 Even in the example shown with the image of the dog and cat, it responds strongly with the corner of the image, which is not interpretable.
Similarly, how does the method respond when the image is cluttered with more information, or the relevant information makes up a much smaller total proportion of the image. More examples are needed.
Choice of layer number, can you justify theoretically why layer 9 has been chosed.

---

> ### Author Response · Authors · 2024-11-17
>
> ### Response to Weaknesses:
> **W1:**
> The identical dimensionality of image and text representations is a design choice inherent to CLIP, intended to facilitate contrastive learning by mapping both modalities into a shared embedding space. Regarding Theorem 1 and Theorem 2, we acknowledge the reviewer’s observation. We will adjust the presentation in the manuscript to describe them as remarks rather than theorems to better align with their fundamental nature.
>
> ### Response to Questions:
> **Q1:**
> We appreciate the reviewer’s request for additional details. Our experiment setup follows the protocol of M2IB, where 2,000 images are used for testing. The computation involves converting boolean values into binary (0 or 1), which limits the precision to fewer decimal places. The robustness of our method is inherently guaranteed by the attribution axioms it satisfies. We have provided additional visual examples in the supplementary material (accessible at [https://anonymous.4open.science/r/NIB-DBCD/rebuttal/] and [https://anonymous.4open.science/r/NIB-DBCD/results/]).
>
> In Section 6.2 of our manuscript, we also present an ablation study on other layers. The rationale for choosing the 9th layer is based on the Bottleneck principle, which hypothesizes the intermediate layer as the most effective bottleneck for representation learning. Through extensive ablation experiments on various datasets, the 9th layer consistently demonstrated optimal performance, balancing detailed feature extraction with high-level semantic representations, but there are also advantages to choosing other layers

---

> > ### Comment · Reviewer_bPyU · 2024-11-26
> > **Reply to authors**
> >
> > Thank you for your comments, in my view your work is of high quality

---

### Meta-Review · Area_Chair_SqqA · 2024-12-20

**Metareview:**

The paper introduces the Narrowing Information Bottleneck Theory (NIBT) as a novel interpretability framework for multimodal image-text representations, addressing critical challenges such as randomness and hyperparameter sensitivity in methods like CLIP and M2IB. NIBT proposes a deterministic mechanism for identifying both positive and negative features in multimodal tasks, ensuring more robust and reliable interpretability. The authors substantiated their claims through experimental validation on datasets and obtained clear improvements in image and text interpretability.

The strengths of this paper lie in its theoretical insights and practical contributions. It provides a novel perspective on interpretability by focusing on negative feature attribution. The method aligns well with modern attribution axioms and eliminates randomness, and is relatively simple to use. The quantitative improvements, combined with detailed ablation studies and public code availability, demonstrate the robustness and reproducibility of the proposed approach.  Most reviewers appreciated the clarity and organization of the manuscript.

However, there are notable weaknesses. The reviewers raised concerns about the lack of diversity in datasets, particularly the absence of high-risk application domains like healthcare. Another concern was the reliance on subjective visualizations, which some reviewers felt lacked specificity compared to competing methods like M2IB. Lastly, minor issues such as formatting inconsistencies in tables and adherence to page limits were flagged and resolved during the rebuttal phase.

I think that this work is overall borderline with both pros and cons. The decision to accept this paper is based on its novelty in theory and method, significant empirical results, and the authors’ engagement during the rebuttal process to address reviewer concerns. After a careful review, I believe that the major concerns from the reviewers have been successfully addressed, and the left ones are less concerning and do not influence the overall decision. In all, I believe that this work would make a valuable contribution. Meanwhile, I suggest the authors take the reviewers' advice carefully and address them in the final revision.

**Additional Comments On Reviewer Discussion:**

The authors propose a detailed response and not all reviewers actively engaged in the discussion. Per the responses from Reviewer bWWD  and Reviewer nFC1, the main concerns in the original response seem addressed. They still have additional concerns preventing higher scores, but overall I think they are less concerning compared to the merits of this work. Given the considerations mentioned above, I appreciate the authors' efforts in rebuttal and recommend acceptance.

---

### Decision · Program_Chairs · 2025-01-22

Accept (Poster)